# Learning efficient task-dependent representations with synaptic plasticity

**Colin Bredenberg**
Center for Neural Science
New York University
cjb617@nyu.edu

**Eero P. Simoncelli**\*
Center for Neural Science,
Howard Hughes Medical Institute
New York University
eero.simoncelli@nyu.edu

**Cristina Savin**\*
Center for Neural Science,
Center for Data Science
New York University
csavin@nyu.edu

## Abstract

Neural populations encode the sensory world imperfectly: their capacity is limited by the number of neurons, availability of metabolic and other biophysical resources, and intrinsic noise. The brain is presumably shaped by these limitations, improving efficiency by discarding some aspects of incoming sensory streams, while preferentially preserving commonly occurring, behaviorally-relevant information. Here we construct a stochastic recurrent neural circuit model that can learn efficient, task-specific sensory codes using a novel form of reward-modulated Hebbian synaptic plasticity. We illustrate the flexibility of the model by training an initially unstructured neural network to solve two different tasks: stimulus estimation, and stimulus discrimination. The network achieves high performance in both tasks by appropriately allocating resources and using its recurrent circuitry to best compensate for different levels of noise. We also show how the interaction between stimulus priors and task structure dictates the emergent network representations.

## 1 Introduction

A variety of forces shape neural representations in the brain. On one side, sensory circuits need to faithfully represent their inputs, in support of the broad range of tasks an animal may need to perform. On the other side, the neural 'wetware' is intrinsically noisy [1], and computing resources are highly limited in terms of the number of neurons and metabolic energy. It remains a mystery how local synaptic learning rules can overcome these limitation to yield robust representations at the circuit level.

Past work has focused on individual aspects of this problem. Studies of efficient coding have successfully explained features of early sensory representations [2, 3] in terms of the interaction between stimulus statistics and resource limitations, and several models have proposed how such representations could emerge through local unsupervised learning [4, 5, 6]. However, the bulk of this theoretical work has ignored task constraints. This oversight might seem justified, considering that we generally think of sensory cortices as performing unsupervised learning, however a growing body of experimental evidence suggests that behavioral goals shape neural receptive fields in adult sensory cortices (A1 [7, 8, 9], S1 [10, 11], V1 [12, 13]), usually in the presence of neuromodulation [14, 15, 16]. This kind of plasticity has been modelled using tri-factor learning rules, which provide a mechanism for learning *task-specific* representations using only local information [17, 18, 19, 20]. However, the interaction between the task, input statistics, and biological constraints remains largely unexplored (but see [21]).

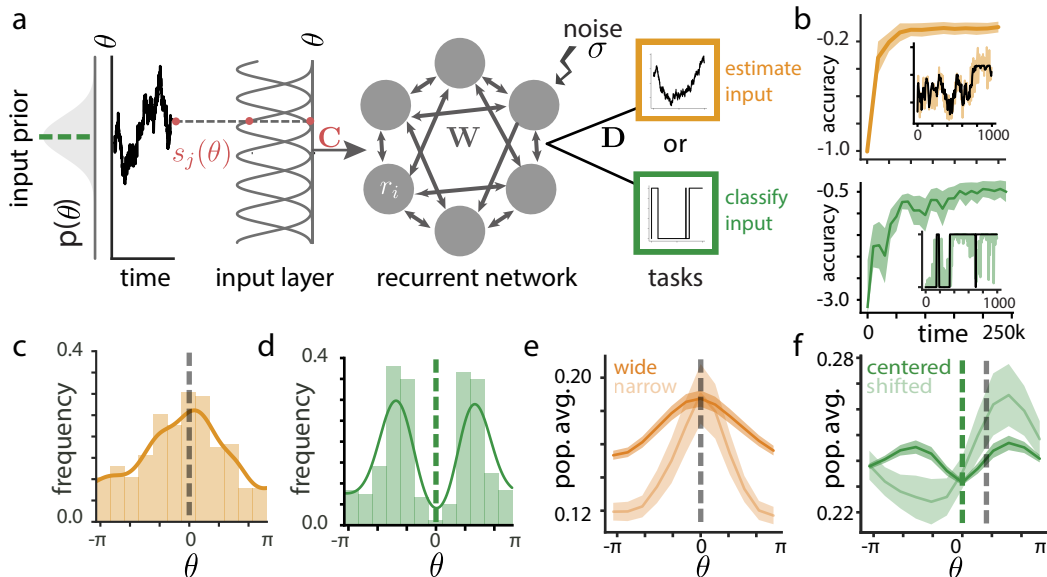

Figure 1: Recurrent neural network architecture and task learning. **a.** Model schematic. Stimuli drawn from a prior distribution (gray) are encoded in the responses of a static input layer, which feeds into the recurrent network; a linear decoder in turn produces the network output for either an estimation or a classification task. **b.** Network performance as a function of training time for the estimation (top) and classification (bottom) tasks. Shaded intervals indicate $\pm$ 1 s.e.m. across 45k test time units. Inset shows an example trained network output, with the correct output in black. **c.** Histogram of preferred stimuli and corresponding kernel density estimates (line). **d.** Same as **c**, for the classification task; decision boundary at $\theta = 0$ (dashed line). **e.** Population-averaged firing rates for two input priors. Shaded intervals are $\pm$ 1 s.e.m. averaging across all neurons in the network (light gold) or all neurons across 20 simulations (dark gold). **f.** Effects of shifting the stimulus prior relative to the classification boundary; dashed lines show the common decision boundary (green) and shifted prior (gray). Shaded intervals as in **e**.

Here we use a stochastic recurrent neural network model to derive a class of tri-factor Hebbian plasticity rules capable of solving a variety of tasks, subject to metabolic constraints. The framework leverages internal noise for estimating gradients of a task-specific objective; thus, noise provides an essential ingredient for learning, and is not simply an impediment for encoding. We systematically examine the interactions between input statistics, task constraints, and resource limitations, and show that the emerging representations select task-relevant features, while preferentially encoding commonly occurring stimuli. The network also learns to reshape its intrinsic noise in a way that reflects prior statistics and task constraints.

## 2 Task-dependent synaptic plasticity

**Stochastic circuit model.** We consider a simple sensory coding paradigm, in which a stimulus orientation $\theta$ is drawn from a von Mises distribution and encoded in the responses of an input population with fixed tuning, $s_j(\theta)$ (Fig. 1a) given by:

$$s_j(\theta) = \lfloor \cos(\theta_j) \cos(\theta) + \sin(\theta_j) \sin(\theta) \rfloor, \tag{1}$$

where $\lfloor \cdot \rfloor$ indicates halfwave rectification, and $\theta_j$ is the maximally-activating stimulus for input unit $j$. This creates a unimodal stimulus response profile, with a peak value of 1 at orientation $\theta = \theta_j$. Further, $\theta_j$ are selected to evenly tile the range $[-\pi, \pi]$.

This activity provides the input to the recurrent network via synapses with weights specified by matrix $\mathbf{C}$. The recurrent network is nonlinear and stochastic, with neuron activities $\mathbf{r}$ and recurrent synaptic strengths $\mathbf{W}$. A linear decoder with synaptic weights parameterized by matrix $\mathbf{D}$ transforms the network activity into a task-specific output.

The stochastic dynamics governing the activity of recurrent neurons take the form:

$$dr_i = \left[ -f^{-1}(r_i) + \sum_{j=1}^{N_r} w_{ij} r_j + \sum_{k=1}^{N_s} c_{ik} s_k + b_i \right] dt + \sigma dB_i, \tag{2}$$

where $f(\cdot)$ is the nonlinear response function (for simplicity, same for all neurons), $N_s$ and $N_r$ are the number of neurons in the input and recurrent populations, respectively, and $b_i$ is a bias reflecting neural excitability. The parameter $\sigma$ controls the standard deviation of the Brownian noise, $B_i$, added independently to each neuron. This intrinsic noise is one of the main constraints on the network. The $f^{-1}(r_i)$ term may seem unusual, but it allows for analytic tractability of the nonlinear dynamics. Furthermore, this formulation allows us to add Brownian noise to the current, such that fluctuations outside of those allowed by the F-I nonlinearity are sharply attenuated. In the small-noise limit, $\sigma \to 0$, the network has the same steady-state dynamics as a traditional nonlinear recurrent neural network. At equilibrium, $\bar{r}_i = f\left( \sum_j w_{ij} \bar{r}_j + \sum_k c_k s_k + b_i \right)$, with nonlinearity $f$ determining the steady-state F-I curve for each neuron; intrinsic noise induces fluctuations about this fixed point.

When the recurrent connectivity matrix $\mathbf{W}$ is symmetric, the network dynamics are a stochastic, continuous analog of the Hopfield network [22, 23], and of the Boltzmann machine [24]. Its corresponding energy function is:

$$E(\mathbf{r}, \mathbf{s}; \mathbf{W}) = -\frac{1}{2} \sum_{i,j} w_{ij} r_i r_j + \sum_i \int_0^{r_i} f^{-1}(x_i) dx_i - \sum_{ij} c_{ij} r_i s_j - \sum_i b_i r_i. \tag{3}$$

The network dynamics implement stochastic gradient descent on this energy, which corresponds to Langevin sampling [25] from the stimulus-dependent steady-state probability distribution:

$$p(\mathbf{r}|\mathbf{s}; \mathbf{W}) = \frac{\exp\left[ -\frac{E(\mathbf{r}, \mathbf{s}; \mathbf{W})}{\sigma^2} \right]}{\int_{\mathbf{r}} \exp\left[ -\frac{E(\mathbf{r}, \mathbf{s}; \mathbf{W})}{\sigma^2} \right]}. \tag{4}$$

The steady-state distribution is in the exponential family, and offers a variety of useful mathematical properties. Most importantly, the probabilistic description of the network, $p(\mathbf{r}|\mathbf{s}; \mathbf{W})$, can be used to calculate the gradient of an objective function with respect to the network weights via a procedure similar to node perturbation [26]. In practice, we use approximate solutions to Eq. (2) (or equivalently, Langevin sampling from Eq.4) using Euler-Maruyama integration.

**Task-dependent objectives.** We consider a task-specific objective function of the general form:

$$\mathcal{O}(\mathbf{W}, \mathbf{D}) = \iint \alpha(\mathbf{Dr}, \mathbf{s}) p(\mathbf{r}|\mathbf{s}; \mathbf{W}) p(\mathbf{s}) \, \mathbf{dr} \mathbf{ds} - \lambda \|\mathbf{W}\|_2^2, \tag{5}$$

where $\alpha(\cdot, \cdot)$ is a task-specific loss function, computed as a function of the linear readout $\mathbf{Dr}$, in a downstream circuit. The second term ensures that synaptic weights do not grow indefinitely [4]; it is a mathematically convenient way of introducing metabolic constraints, although regularizing the neural activity itself is also possible. For brevity, we have only included the constraint on $\mathbf{W}$ in Eq. (5). In practice, we also include similar constraints on $\mathbf{C}$, $\mathbf{D}$, and $\mathbf{b}$, with corresponding Lagrange multipliers $\lambda_C$, $\lambda_D$ and $\lambda_b$.

The specific choices of the loss, $\alpha$, determines the nature of the task. Here, we chose two example objective functions – input encoding and binary classification. For reproducing the input, we use a mean squared error (MSE) objective:

$$\alpha_{\text{MSE}}(\mathbf{Dr}, \mathbf{s}) = -\|\mathbf{s} - \mathbf{Dr}\|_2^2, \tag{6}$$

with a negative sign reflecting the fact that we are maximizing, rather than minimizing, the objective.

For classification, we use a cross-entropy objective:

$$\alpha_{\text{LL}}(\mathbf{Dr}, \mathbf{s}) = \phi(\mathbf{s}) \log(\psi(\mathbf{Dr})) + (1 - \phi(\mathbf{s})) \log(1 - \psi(\mathbf{Dr})), \tag{7}$$

where $\psi(\cdot)$ is a sigmoid nonlinearity and $\phi(\mathbf{s})$ gives the mapping from stimulus $\mathbf{s}$ to corresponding binary class.

**Local task-dependent learning.** We derive synaptic plasticity rules by maximizing $\mathcal{O}(\mathbf{r}, \mathbf{s}; \mathbf{W})$ using gradient ascent, averaging across the stimulus distribution $p(\mathbf{s})$, and taking advantage of the closed-form expression for the steady-state stimulus-dependent response distribution, $p(\mathbf{r}|\mathbf{s}; \mathbf{W})$.

Taking the derivative of the objective function (Eq. 5) with respect to $w_{ij}$ yields:

$$\frac{\partial \mathcal{O}}{\partial w_{ij}} = \iint p\left(\mathbf{s}\right) \alpha\left(\mathbf{Dr}, \mathbf{s}\right) \frac{\partial p\left(\mathbf{r}|\mathbf{s}\right)}{\partial w_{ij}} \mathbf{dr}\mathbf{ds} - 2\lambda w_{ij}. \tag{8}$$

As in [27], differentiating Eq. 4, we note that:

$$\frac{\partial}{\partial w_{ij}} p\left(\mathbf{r}|\mathbf{s}; \mathbf{W}\right) = \frac{-1}{\sigma^2}\left[\frac{\partial}{\partial w_{ij}} E(\mathbf{r}, \mathbf{s}; \mathbf{W}) - \left\langle \frac{\partial}{\partial w_{ij}} E\left(\mathbf{r}, \mathbf{s}; \mathbf{W}\right)\right\rangle_{p(\mathbf{r}|\mathbf{s})}\right] p\left(\mathbf{r}|\mathbf{s}; \mathbf{W}\right), \tag{9}$$

where the brackets denote the conditional expectation with respect to $p(\mathbf{r}|\mathbf{s}; \mathbf{W})$. Rearranging, and substituting Eq. (9) into Eq. (8) yields:

$$\frac{\partial \mathcal{O}}{\partial w_{ij}} = \frac{-1}{\sigma^2} \iint \alpha\left(\mathbf{Dr}, \mathbf{s}\right) \left(\frac{\partial E\left(\mathbf{r}, \mathbf{s}; \mathbf{W}\right)}{\partial w_{ij}} - \left\langle \frac{\partial E(\mathbf{r}, \mathbf{s}; \mathbf{W})}{\partial w_{ij}}\right\rangle_{p(\mathbf{r}|\mathbf{s})}\right) p(\mathbf{r}, \mathbf{s}; \mathbf{W}) \, \mathbf{dr}\mathbf{ds} - 2\lambda w_{ij}$$

$$= \frac{1}{\sigma^2} \iint \alpha(\mathbf{Dr}, \mathbf{s}) \left(r_i r_j - \langle r_i r_j\rangle_{p(\mathbf{r}|\mathbf{s})}\right) p(\mathbf{r}, \mathbf{s}; \mathbf{W}) \mathbf{dr}\mathbf{ds} - 2\lambda w_{ij}.$$

To update weights via gradient ascent, the learning rule takes the form:

$$\Delta w_{ij} \propto \frac{\partial \mathcal{O}}{\partial w_{ij}} = \mathbb{E}\left[\alpha(\mathbf{Dr}, \mathbf{s})\left(r_i r_j - \langle r_i r_j\rangle_{p(\mathbf{r}|\mathbf{s})}\right)\right] - 2\lambda_W w_{ij},$$

where we have assigned $\lambda_W = \sigma^2 \lambda$. We further approximate the expectation by sampling $\mathbf{r}$ as part of the network dynamics and update weights according to a single sample (if $\Delta w_{ij}$ is sufficiently small, this is equivalent to updating $w_{ij}$ with an average over several samples):

$$\Delta w_{ij} \propto \alpha(\mathbf{Dr}, \mathbf{s})\left(r_i r_j - \langle r_i r_j\rangle_{p(\mathbf{r}|\mathbf{s})}\right) - 2\lambda_W w_{ij}. \tag{10}$$

This expression for the weight update is similar to a standard reward-modulated Hebbian plasticity rule. It is driven by correlations between pre- and post-synaptic activity, with reward, $\alpha$, having a multiplicative effect on weight changes. The subtractive term ensures that updates only occur in the presence of deviations from the average correlation level. It is symmetric in the indices $\{i, j\}$, and thus preserves the symmetry of the weight matrix $\mathbf{W}$. Finally, the weight regularization adds a contribution similar in form to Oja's rule [4]. In practice, we approximate the conditional expectation by a running average computed using a low-pass filter. We can derive similar updates for the input weights, and biases:

$$\Delta c_{ik} \propto \alpha(\mathbf{Dr}, \mathbf{s})\left(r_i s_k - \langle r_i s_k\rangle_{p(\mathbf{r}|\mathbf{s})}\right) - 2\lambda_C c_{ik}$$

$$\Delta b_i \propto \alpha(\mathbf{Dr}, \mathbf{s})\left(r_i - \langle r_i\rangle_{p(\mathbf{r}|\mathbf{s})}\right) - 2\lambda_b b_i,$$

Hence, our framework allows us to optimize parameters of the network using local operations.

It is worth comparing our plasticity rule to REINFORCE [27], which would have, for a discrete-time RNN: $\Delta w_{ij} \propto \alpha(\mathbf{Dr}, \mathbf{s}) \sum_{t=0}^{T} f'(h_i(t))(r_i(t) - \bar{r}_i(t))r_j(t) - 2\lambda_W w_{ij}$, where $h_i$ is the pre-activation of neuron $i$, and $\bar{r}_i(t)$ is the expected average activation for that neuron at time $t$. One notable difference is that learning is gated by deviations from the mean co-activation of the pre- and post-synaptic neurons, and not by post-synaptic activity alone. Another difference is the presence of a sum over time (an eligibility trace), which is not required for our plasticity rule. Lastly, this rule includes an $f'(h_i(t))$ term, whereas ours only involves the firing rates of the neurons. At least some of these differences could be examined experimentally.

**Learning the decoder.** Because the readout weights enter $\alpha(\mathbf{Dr}, \mathbf{s})$, the optimization of $D$ requires a slightly different treatment. Since $p(\mathbf{r}|\mathbf{s}; \mathbf{W})$ does not depend on $\mathbf{D}$, taking the derivative of Eq. (5) yields:

$$\frac{\partial \mathcal{O}}{\partial D_{ij}} = \int p(\mathbf{s}, \mathbf{r}; \mathbf{W})\frac{\partial}{\partial D_{ij}} \alpha(\mathbf{Dr}, \mathbf{s}) \, \mathbf{dr}\mathbf{ds} - 2\lambda_D D_{ij}.$$

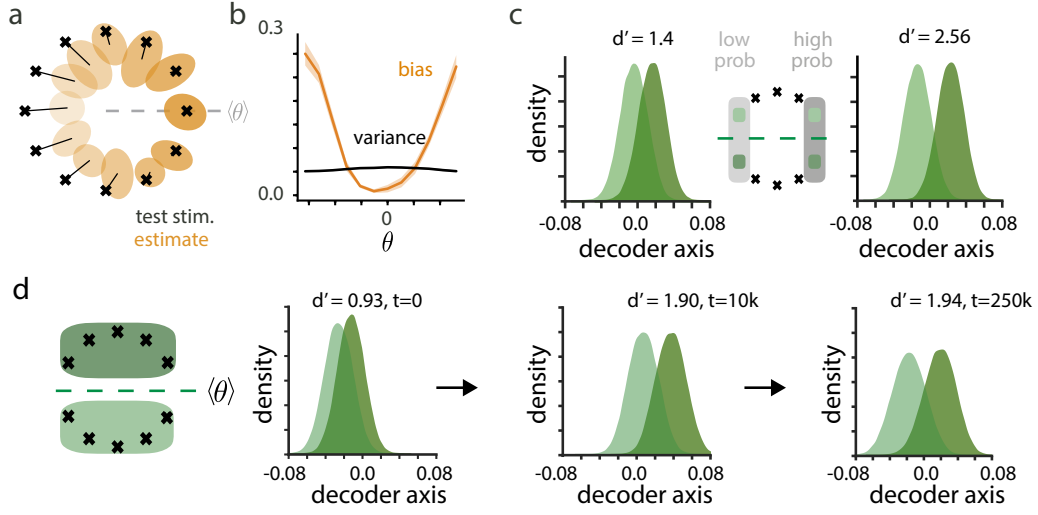

Figure 2: Task-specific stimulus encoding. **a.** Outputs for the estimation task: crosses mark the locations of test stimuli; black lines indicate the bias, or the distance between the target and the mean output; ellipses show the 95% probability region for the associated network response distributions. Lighter colors are further from the most probable stimulus ($\langle\theta\rangle = 0$). **b.** Squared bias and variance for the estimator task as a function of stimulus angle. Shaded intervals are $\pm$ 1 s.e.m. across 20 simulations. **c.** Decoder response distribution for stimuli near the boundary in the low-probability region of the space ($\theta = \pi$, left), and in the high-probability region ($\theta = 0$, right), as indicated on the center schematic. **d.** Left: network output schematic for test stimuli (black crosses). The green patches indicate the two target classes, and the green dashed line indicates the classification boundary. $\langle\theta\rangle$ indicates the highest probability stimulus. Right: discriminability of stimulus classes in the network output, measured by the sensitivity index ($d'$), shown before learning, after 10k time units, and after 250k time units.

Using the same stochastic update scheme as in Eq. (10) yields:

$$\Delta D_{ij} \propto \frac{\partial}{\partial D_{ij}}\alpha(\mathbf{Dr}, \mathbf{s}) - 2\lambda_D D_{ij}.$$

This partial derivative will be different for each choice of $\alpha$. For $\alpha_{\mathrm{MSE}}$, we get:

$$\Delta D_{ij}^{(\mathrm{MSE})} \propto 2\left(\sum_k D_{ik}r_k - s_i\right)r_j - 2\lambda_D D_{ij}, \tag{11}$$

and for $\alpha_{\mathrm{LL}}$ (noting that $\mathbf{D}$ here has only one row):

$$\Delta D_j^{(\mathrm{LL})} \propto (\phi(\mathbf{s}) - \psi(\mathbf{Dr}))\,r_j - 2\lambda_D D_j. \tag{12}$$

## 3 Numerical results

To simulate the realistic scenario of slowly changing input stimuli constantly driving the circuit, we sample inputs from a von Mises prior distribution using Langevin dynamics with a significantly slower time constant than that of the recurrent network dynamics, given by $\tau_s = 375$. We set the noise level to an intermediate regime, so that its effects on circuit computation are non-negligible, but not pathological ($\sigma = 0.2$), and calibrate the hyperparameters that control the metabolic constraints (strength of regularization) to a level where they start to interfere with network performance.[1] As learning progresses, our derived local plasticity rules quickly converge to a good solution, for both estimation and categorization (Fig. 1 b).

The emerging representations are noticeably different for the two tasks (compare Fig. 1c and d): for estimation, the distribution of preferred orientations is concentrated in the highly probable stimulus region, whereas the preferred distribution is bimodal in the case of classification. The average population activity for any stimulus provides additional quantification of the way in which learning allocates network resources (here, neural activity) to different stimuli (light colors in Fig. 1e and f). For the representation task, this metric confirms that neural resources are preferentially allocated to commonly occurring stimuli (Suppl. Fig. S1a, Fig. 1c; the most likely stimulus is 0). Hence learning has converged to an efficient representation of stimulus statistics. Moreover, the average population activity encodes the prior probability of input stimuli (Fig. 1e).

The picture looks very different in the case of categorization: improbable stimuli ($\theta = \pm\pi$) still have a small contribution to the emerging neural representation, but so do the most probable stimuli (which in our example are centered on the categorization boundary). This is reflected in the distribution of preferred stimuli: the neurons distribute their tuning functions on either side of the most probable stimulus so that the most sensitive part of their tuning function lies on the decision boundary (Suppl. Fig. S1b), and their peak responses are tuned for class identity (Fig. 1d). Overall, the emerging representations reflect both input statistics and task constraints.

So far all results used a particular choice of prior, but the same intuitions hold under prior variations. In the estimation task, training a new network with a tightened prior leads to a corresponding tightening of the population tuning (Fig. 1e, Suppl. Fig. S1c). Firing rates decrease for peripheral stimuli; under synaptic weight regularization, the network will reduce firing rates for less probable stimuli, as they have little impact on the average error. This firing rate reduction is coupled with a concomitant increase in error for less probable stimuli (Suppl. Fig. S1d). For the classification task, we trained a new network with a shifted input distribution (Suppl. Fig. S1e) as a way to segregate the effects of the decision boundaries from those of the prior distribution Fig. 1f). This break in symmetry leads to asymmetric errors, with larger errors for the less probable class (Suppl. Fig. S1f), though the effect is more subtle than that observed when tightening the input distribution. The corresponding network representation also becomes asymmetric, with higher firing rates concentrating on the side of the more probable stimulus. Thus, even under transformations of our original prior, trained neurons show higher firing rates for both more probable, and more task-relevant features of the input, such that performance is consistently better for frequent stimuli.

How do these task-specific changes in representation manifest themselves in the decoded outputs? First, in the case of estimation, we can probe the distribution of outputs of the linear readout (Fig. 2a, gold ellipses) for a set of test stimuli (Fig. 2a, black crosses).[2] We found that responses are systematically biased for less probable stimuli, whereas the bias is negligible for frequently occurring stimuli. The effect on variance is much weaker (Fig. 2b). Second, for classification, we need to measure decoder output variability as a function of the prior stimulus probability. We use the fact that the classification boundary intersects the circle on which the input stimuli lie for the most probable and the least probable stimuli under the prior (Fig. 2c, inset). We compare the degree of output overlap for two test stimuli equally spaced on the two sides of the decision boundary at these two extremes. We find substantially higher discriminability for the high probability stimuli, relative to the low probability ones. As for estimation, this difference is due primarily to a better separability of the two means rather than a difference in variance. In summary, the network exhibits better performance for probable stimuli across tasks. Given limited resources, input statistics dictate not only the precision of the representation, but also task performance.

Next, we investigate the dynamics of learning and associated network representations, focusing on the classification task. We measure output discriminability at different stages of learning, with sensitivity index $d'$ values estimated averaging over the stimulus distribution. At the outset, the output distributions for the two stimulus categories are indistinguishable (Fig. 2d). As learning progresses, the means of the two distributions segregate, while their variance remains approximately the same; this result prompts a more detailed examination of the degree to which the network is able to compensate for its own intrinsic noise.

Intrinsic noise in the recurrent dynamics is a key component of our solution, because the learning rule changes synaptic strengths based on whether fluctuations in the synapses' pre- and post-synaptic activities are associated with an increase or decrease in performance (Eq. 10). However, adding

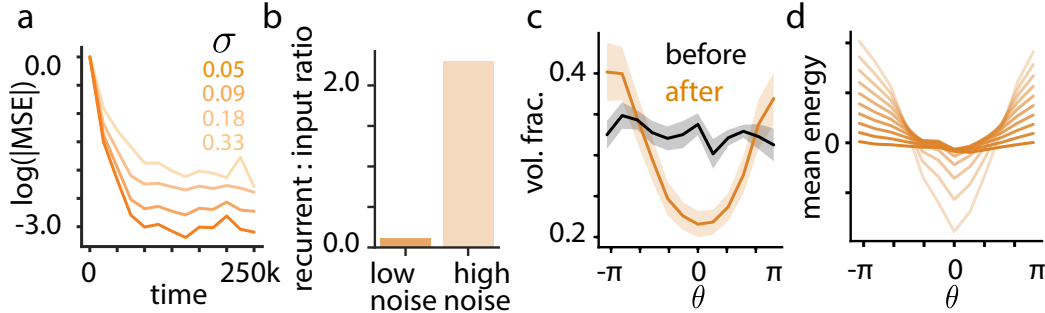

Figure 3: The effects of internal noise. **a.** Estimation performance during learning for different magnitudes of noise $\sigma$ (for reference, the average external drive to the neuron is roughly twice the size of the largest noise level). **b.** Ratio of recurrent to input current for neurons in a low vs. high noise regime. **c.** Volume fraction of noise within/outside the readout manifold, as a function of stimulus angle, before and after learning ($\sigma = 0.33$). Shaded intervals are $\pm\ 1$ s.e.m. across 20 simulations. **d.** Mean energy of $p(\mathbf{r}|\mathbf{s})$ as a function of input angle and different magnitudes of noise $\sigma$.

noise to the network can also make estimation more difficult: we found that increasing intrinsic noise leads to both slower learning and worse asymptotic performance (Fig. 3a). Given that output variance changes little across stimuli and over learning, is noise strictly deleterious for the network, and how can the network learn to counteract its effects? Interestingly, the more intrinsic noise we add in the network, the more the network relies on recurrent connectivity for its representation. Increases in noise cause an increased engagement of recurrent connectivity after learning (Fig. 3b). The fact that recurrent connectivity is not strictly needed in absence of noise is intuitive, given that our chosen tasks involve a simple, instantaneous map of inputs to outputs (although note that in our framework some noise is needed to enable synaptic plasticity to occur). Though it is clear that recurrent connections alleviate some of the negative consequences of noise, their exact mechanism of action requires more investigation.

It seems like the overall level of noise in the network does not change dramatically during learning, at least not as reflected in output fluctuations. Yet, performance is systematically better for probable stimuli, indicating the possibility of noise compensation. To investigate whether (and if so, how) recurrent connections shape internal noise on the estimation task, we asked what fraction of the internal noise lies in the decoding manifold given by $\mathbf{D}$ and if it depends on the stimulus. Since the entropy of the network response distribution conditioned on the stimulus and its marginals are not analytically tractable, we resorted to numerical approximations: we used the network dynamics (with frozen weights) to sample from this conditional distribution, projected these responses a) in the readout manifold and b) in the 2D manifold of maximum variance, as defined by the first two PCA axes of the neural responses. Our final metric, which we refer to as the noise volume fraction, is computed as the ratio between the estimated noise in each of the two manifolds (using the determinant of the empirical covariance of the projected responses as a proxy for noise magnitude). The noise volume fraction is defined as follows:

$$VF(\mathbf{s}) = \frac{\det \mathbf{C}_D}{\det \mathbf{C}_{\text{PCA}}}, \tag{13}$$

where $\mathbf{C}_D$ is the covariance matrix of $\mathbf{r}$ projected onto the two output dimensions, and $\mathbf{C}_{\text{PCA}}$ is the covariance matrix of $\mathbf{r}$ projected onto the first two principal components of the neural activity for a fixed stimulus $\mathbf{s}$. This metric is 1 when the primary axes of internal noise lie in the decoding manifold; it is 0 when the two spaces are orthogonal, such that internal noise does not affect network outputs. After learning, the volume fraction is much smaller for probable stimuli (Fig. 3c), indicating that the network has learned to effectively 'hide' more of its noise for frequent inputs. This is not to say that output variance is lower for more probable stimuli: as we have already seen, if anything, output variance increases slightly for more probable stimuli. In general, the variability of the network output increases with the firing rates of its neurons, such that high network activity *necessarily* produces increased variability (one cannot amplify the signal without amplifying the noise to some degree). However, when normalizing for this increase, the network projects a smaller fraction of its total noise onto the decoder for more probable stimuli than it does for less probable stimuli.

An alternative way to think about the effects of intrinsic noise on the network activity is in terms of the energy function (Eq. 3), and corresponding steady-state stimulus response distribution (Eq. 4). Here, the noise variance acts as a temperature: the energy landscape flattens with increasing noise. Formally, one way to compensate for a increases in $\sigma$ is to rescale the the network's stimulus-conditioned energy in proportion to the increase in $\sigma^2$, thus preserving the stimulus response distribution $p(\mathbf{r}|\mathbf{s})$. The network does employ this kind of compensation (Fig.3d). As the intrinsic noise increases, the network boosts the gain of its energy function, with the mean energy increasing monotonically as a function of $\sigma^2$ (Suppl. Fig. S1g). Hence, the network learns to compensate for its intrinsic noise so as maintain a good signal to noise ratio.

## 4  Discussion

Despite recent successes in the supervised training of recurrent networks [28, 29, 30, 31], it remains a mystery how biological circuits are capable of robust, flexible local learning in the face of constraints such as internal noise and limited metabolic resources. Here we have derived a class of synaptic learning rules that optimize a task-specific objective by combining local information about the firing rates of each pair of connected neurons with a single performance signal, implementable by global neuromodulation. Online learning naturally follows, since the network dynamics sample from a well-defined stimulus response distribution.[3] We further show that the derived learning rules lead to emerging neural representations that allocate neural resources and reshape internal noise to reflect both input statistics and task constraints.

The use of stochasticity as a means of estimating gradients has an extensive history [27] and has been used to account for biological phenomena, in particular the role of variability of neurons in the songbird HVC nucleus in song learning [26]. Our model is conceptually similar, and can be thought of as a mathematically tractable instantiation of the REINFORCE framework [27], where stochastic network responses are given by a continuous variant of the Boltzmann machine [24]. Our model notably lacks an eligibility trace, which in previous models was required to integrate coactivity through time at synapses; it also stores an averaged measure of coactivity, $\langle r_i r_j \rangle_{p(\mathbf{r}|\mathbf{s})}$, rather than only averaging over post-synaptic activity–both of these differences are potentially experimentally testable. Further, existing literature has focused exclusively on the role of intrinsic noise on learning dynamics, ignoring its interactions with the circuit function and the emerging neural representations that support it. Here we show that there is a conflict between the positive role of stochasticity on learning and its deleterious effects on encoding. Over the course of learning, the network converges to a solution that trades off between the two, by appropriately reshaping the noise so as to increase the signal-to-noise ratio of the output.

Prior statistics play a key role in the emerging representations, with more neurons tuned to commonly occurring stimuli, and overall population activity weaker for infrequent stimuli. The inhomogeneous distribution of tuning functions aligns with optimal encoding of priors, as derived for abstract encoding models [33]. But unlike previous models, our network also encodes the prior as a population tuning function, a discrepancy that most likely reflects differences in the exact form of the regularizer enforcing metabolic constraints. Different choices of regularizer, in particular a sparseness-encouraging constraint on neural activity, would likely lead to representations more similar to traditional efficient coding models [2]. Nonetheless, our approach provides explicit local learning dynamics for these abstract models and, importantly, is successful in regimes where analytic methods are intractable, e.g. for multivariate stimuli, and for tasks that go beyond simple stimulus reproduction.

One limitation of our formulation is that reward does not depend on the history of network responses. This differentiates our approach from traditional models of reward-modulated learning, which focus on solving temporal credit assignment, especially in spiking circuits [18, 26, 34, 35]. Despite this limitation, our mathematical derivation extends previous work by making explicit the four-way interactions between intrinsic noise, metabolic constraints, input statistics, and task structure in the circuit. Preliminary results suggest that it is possible to extend the current framework to incorporate temporal dependencies in both stimuli and reward structure, better aligning it with traditional goals of reinforcement learning.

It has been argued that learning algorithms based on stochastic gradient estimates cannot match the learning capabilities of the brain [36], as they perform poorly in high dimensions [37]. This has lead to renewed focus on alternatives that rely on the neural system having access to a closed-form expression for the gradient (at least approximately), in particular biologically-plausible approximations to backpropagation [38, 36, 31]. Still, these models can't be the whole story. While gradient information might be available for unsupervised [39] or intrinsic learning objectives [40], this is certainly not true for external rewards, when the loss function is specified by the environment itself. Animals rarely, if ever, have access to explicit reward functions. Moreover, neither unsupervised learning nor backpropagation can account for the critical role of neuromodulation in synaptic plasticity and its contribution to perceptual learning [14, 15, 16]. The complementary nature of the two classes of learning rules suggests that they might *both* play an important role in biological learning. Bringing the two closer together is a promising direction for future research, both theoretical and experimental.

## 5    Broader Impact

Here we develop a theoretical framework for the neural substrates of perceptual learning in animals and humans. The moral valence of this work is largely determined by its application: understanding perceptual learning could potentially have a beneficial impact by leading to a better theory of developmental disorders, such as amblyopia, autism, or schizophrenia, but it could also have a potentially negative impact by being used as a tool to manipulate learning in consenting individuals. As our work is theoretical, biases caused by data, and potential system failures are not applicable.

## Acknowledgments and Disclosure of Funding

We thank the following funding sources for their support: NRT-HDR 1922658, a Google faculty award (CS), and the Howard Hughes Medical Institute (ES, CB). The authors declare no competing interests.

## Footnotes

\*indicates equal contribution.

[1]For parameter values, see https://github.com/colinbredenberg/Efficient-Plasticity-Camera-Ready.

[2]Plasticity was disabled for test stimuli here, to isolate effects of internal noise without the confound of changes in network parameters.

[3]Note that although we are using sampling dynamics as well, our approach is different from sampling theory [32] in that here the sampling dynamics do not represent a Bayesian posterior.

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
