[Supplementary Material]

# Learning efficient task-dependent representations with synaptic plasticity –Supplementary Information–

**Colin Bredenberg**
Center for Neural Science
New York University
cjb617@nyu.edu

**Eero P. Simoncelli***
Center for Neural Science,
Howard Hughes Medical Institute
New York University
eero.simoncelli@nyu.edu

**Cristina Savin***
Center for Neural Science,
Center for Data Science
New York University
csavin@nyu.edu

Figure S1: Manipulations of the input distribution. **a.** Sample tuning curves for the representation network. The gray dashed line indicates the prior mode. **b.** Sample tuning curves for the classification network. The green dashed line indicates the prior mode and the classification boundary. **c.** Two different input priors: the original distribution in light gold, and a sharper prior in dark gold. **d.** Error as a function of test input angle for a network trained on the original distribution (light gold), and the tightened distribution (dark gold). **e.** Schematic showing shifting the input distribution for the classification task. The original distribution is shown in light green, and the shifted distribution is shown in dark green; the green dashed line indicates the classification boundary. **f.** Error as a function of test input angle for a network trained on the original distribution (light green), and the shifted distribution (dark green). The gray dashed line indicates the shifted input distribution mode. Error bars for the original distribution indicate +/- 1 s.e.m. across 20 simulations. **g.** The gain (L2 norm) of the mean energy as a function of $\sigma^2$.

*indicates equal contribution.