[Reviews · NeurIPS 2020]

Review 1

Summary and Contributions: This paper proposes a stochastic recurrent neural network that builds up its local information representation through a learning rule based on Boltzmann machines, but weighted by a task-dependent objective function, forming a so-called tri-factor learning rule. The results show how the network depends on the tasks of regression and classification in terms of the distribution of the tuning curves, population-averaged activities, and dependence on stimulus priors. The paper then considered how noises are redistributed in the neural manifold such that task performance can be achieved.

Strengths: This paper has several strong points. The first one is its thoroughness in elucidating the properties of the learning rule it introduced. Besides merely showing that it works, the paper discusses how the network behaviors such as the distribution of the tuning curves depend on factors such as the choice of the task and the extent of congruence between the most probable input and the decision boundary. The focus of the paper is comprehensive in its attention paid to the behavior at the neuronal level as well as the population level. Another strong point is the discussion on the tuning curves of the neurons. This enables the comparison of its predictions and related models with neuroscience experimental results. The discussion on the effects of prior shifts and discriminability are also relevant to the neuroscience community. One further contribution of the paper is the study of the effects of learning on noise redistribution through its introduction of the notion of noise volume fraction. This concept and technique can be extended to the study of other models.

Weaknesses: Despite the theoretical insights, many details were not written clearly. Please see comments in the section on clarity. Update after feedback and discussions: ============================= The main contribution of this paper is the insightful elucidation of the learning behaviors of the tri-factor learning rules. The authors’ response on the symmetric weights is less satisfactory. If the weights are not symmetric, it is not possible to write an energy function to be minimized in the first place, and an alternative formulation is needed. I accept the authors’ clarification of the volume fraction. On the other hand, the argument that noises were introduced to derive the closed-form expressions and to avoid trapped learning was in fact already well known. The original sentence might have misguided the reader to think of deeper principles not intended by the authors. Related to the issue of noises in the authors’ response to Reviewer 3, I agree that more efforts should be made to consider how stochasticity encodes uncertainty, rather than merely a tool to derive convenient algorithms. For example, variances of neural responses play a role in encoding uncertainty in probabilistic population coding.

Correctness: The claims are fine, but restricted to the case of symmetric connectivity matrices, for which the energy function can be written and gradient descent can be derived. The authors may like to discuss how the situation would be modified for asymmetric couplings.

Clarity: Surely there is room for improvement. Below are some examples: Fig. 1 caption: “Kernel density estimates” is not explained. Line 43: The fixed tuning $s_j(\theta)$ is not explained. This is an important point since the tuning profile will affect the output tuning curves. Eq. (8): A minus sign is missing. Line 110: Where is the symbol $\tau_s$ defined? Line 112: Is the symbol $\sigma_B$ the same as $\sigma$? Line 126: It was asserted that contribution of improbable stimuli to the emerging representation is small. The phrase “improbable stimuli” is ambiguous, since it may refer to either $\theta = \pi/2$ where the frequency in Fig. 1d is highest, or to the extreme values $\theta = \pm \pi$. The reader will get confused if he/she has the former case in mind. Fig. 2a: Explanation to this figure is missing. For example, what does the shapes and colors of the ellipses represent, and what do the pins connecting the ellipses and the crosses represent? Fig. 2d: Similarly, explanation to this figure is missing. For example, what do the patches represent? What do their shapes and colors represent? Lines 132-144: In the experiments with changed priors, it is not clear to the reader whether the priors are changed during learning or only during their testing, or the simulation is performed with online learning with simultaneous online testing. Line 160 and Fig. 2: The symbol $d’$ is not explained. Line 180: The conditional stimulus was mentioned, but it is not clear the stimulus is conditional of what. Line 185: The noise volume fraction was introduced, but no mathematical formulation was presented in either the main text or Supplementary Material. If this is an important conclusion of the paper, the reader deserves to know more details. Line 213: It is not clear in what way “intrinsic noise in the recurrent dynamics ... allows us to derive closed-form probabilistic expressions for the objective function gradients.” If it refers to the factor $1/\sigma^2$ in Eq. (8) which was discarded in Eq. (9), the reasoning is not strong enough as this is merely an issue of scaling. Ref [32]: The volume and pages are missing.

Relation to Prior Work: A few references on tri-factor learning were mentioned and it was pointed out that many details remained unexplored. The authors may also like to note the following reference: Lukasz Kusmierz, Takuya Isomura and Taro Toyoizumi, Learning with three factors: modulating Hebbian plasticity with errors, Current Opinion in Neurobiology 46:170-177 (2017).

Reproducibility: No

Additional Feedback: Would the authors like to consider more complex tasks as the next step?


Review 2

Summary and Contributions: In this paper the authors derive an error modulated hebbian learning rule and show how it can be used in a recurrent neural network to learn an estimation or classification task. They then investigate how the learned representations vary across these two tasks.

Strengths: The authors' investigation of the difference in neural tuning, and how these are shaped by the different priors and loss functions between tasks, was thorough.

Weaknesses: It's unclear what the major contributions of the current paper are, when compared to the cited literature. Specifically, the derivation of learning rules (eq 4-9) results in equations nearly identical to the previous literature (eg: eq 7 of citation 34, eq B3 25, eq 16 of Legenstein, Chase, Schwartz, Maass 2010), once the conditional expectations are replaced with time-averaged values. The introduction of two different read out terms (eq 5 & 6) represents only incremental progress by introducing the ability to perform two different types of tasks.

Correctness: The methodology and interpretation of results seem correct.

Clarity: The paper is written clearly, and the presentation doesn't inhibit comprehension.

Relation to Prior Work: It's unclear how the proposed learning rule is different from the cited works.

Reproducibility: Yes

Additional Feedback: [Following author feedback]: It's still unclear to me what the major novelty of the derived learning rule is. As the authors point out in their rebuttal the learning rule takes the form of a modulation term multiplied by the difference between short term and longterm pre-post correlations. This is the form taken in standard Contrastive Hebbian Learning, with an additional modulating term. However, I will defer to my fellow reviewers in the novelty and contributions of having derived the learning rule in a principled manner.


Review 3

Summary and Contributions: This study uses a normative approach to derive a learning rule in a network model with consideration of task information. The derived learning rule resembles the reward-modulated Hebbian rule, as claimed by the author. To illustrate the model, the author used this model to solve two tasks with estimating continuous and categorical variables.

Strengths: The mathematical analysis in this work is solid, and the simulated experiments support their claims.

Weaknesses: I have a couple of major concerns on conceptual levels. I appreciate if the authors could add related discussions in a revised version (there is still enough space in Discussions) or briefly answer them in the author feedback. 1. I am happy to see the results that how different tasks influence the representation in a recurrent network. However, I am wondering the possibility of a disentangled representation of sensory information and the task information. That is, the learned recurrent weight W only depends on the sensory information (likelihood and prior) but not on the task information, while the task information is uniquely represented on the readout matrix D. An advantage of this representation is that the brain doesn’t need two sets of recurrent weight W to implement two different tasks, while the brain only needs to choose how to readout the sensory information represented in the network. Update after rebuttal: the authors cite two papers support a mixed representation between sensory and task information. I suggest the authors add a brief discussion of this in the revised manuscript. This could also enhance the biological relevance of this work. 2. Although the results of current study imply there is a conflict between the roles of stochasticity on learning and on encoding (line 225), I am hesitate to accept this claim in general. I think it is probably not true if the authors consider a Bayesian framework where the posterior distribution is represented by sampling-based codes. In this framework, the internal stochasticity is also essential to encode the posterior distribution, but not harmful for encoding. Line 225: conflict between the positive role of stochasticity on learning and its deleterious effects on encoding. The conflict may not be true if you consider the sampling-based representation of posterior distribution in a Bayesian framework. The network just plays as a filter of the noise in the framework of MSE (point estimate).

Correctness: I have gone through all math equations and I believe the math derivations are correct although some typos exist (see the comments below). The numerical experiments support the conclusions.

Clarity: Overall the paper is well written and I could get the main information quickly. Some suggestions on the structure and typos: 1. Although the authors mentioned the stochastic network is performing sampling, I have been puzzled that why they only consider objective functions based on point estimate (Eqs. 5-6) rather than whole posterior distribution. My puzzled was not relieved until I saw the footnote on page 7. I strongly suggest the authors mention this at somewhere earlier, e.g., when they mention sampling and objective functions on pages 3 and 4. 2. Line 40: I think the title of “local learning” is a little bit over-claimed because the learning rule also depends on a global task-specific loss function. Although I do accept the derived learning rule and its biological plausibility, I suggest the author revise this title. 3. Eq. 6: I think the objective function is the cross entropy if I understood correctly. And then the term inside the 2nd log function should be 1- \psi. 4. For the Eq. right after line 90: does it lose 1/\sigma^2 for the first term on the right-hand side? 5. Eq. 9: compare with the Eq. right after line 90, I think you cannot just throw away the expectation when approximating it by sampling. Some standard notation is replacing the expectation by an empirical sum over samples (see Eq. 11.2 in Bishop 2006’s book for an example). For example, you could write Eq. 9 as the empirical average over neuronal responses r which has an index t of time. 6. Line 220: what is the full name of HVC? Is it a brain area or a nucleus in a songbird? Even if it is a common and standard abbreviation in songbird study, briefly mentioning what this means is quite helpful for readers.

Relation to Prior Work: The author discussed related work on stochasticity, learning rules, etc.

Reproducibility: Yes

Additional Feedback:


Review 4

Summary and Contributions: The authors detail a framework for stochastic gradient estimation of task-based objective functions. They demonstrate its application in a noisy recurrent neural network design in which the stochastic gradient updates can be expressed as three-factor Hebbian update rules for the parameters. Experiments show that the authors’ network learns to “allocate” noise in a way that helps it maximize task performance and represent common stimuli well.

Strengths: The authors clearly situate their work within the realm of bioplausible gradient-estimation approaches, and give a very clear exposition. I know this will sound cliche, but after reading their paper I think, “aha, if I had read the background material, I could have come up with this!” While I am partial to a Bayesian approach, their derivations make the paper worth reading before I reach the numerical results.

Weaknesses: The authors could have used an experimental design that would be easier to scale up and evaluate from a machine learning perspective, or from the perspective of broader perceptual neuroscience. Real perception involves multiple receptors, modalities, etc. On the other hand, datasets with easy-to-evaluate probability distributions for these kinds of stimuli are harder to come by. I would also like to have seen the authors discuss a broader class of neural network designs, ideally separating the network architecture itself from the Hebbian gradient ascent algorithm. After all, we do not know what sort of network architecture the brain itself uses, so an algorithm or update rule derivation that can apply to arbitrary architectures would seem, at least to me, to be more biologically plausible than one that constrains us to fully connected recurrent networks.

Correctness: I cannot find any clear mistakes in their application of the REINFORCE gradient estimation trick, nor in the plots of their numerical results.

Clarity: The paper is very clearly written, one of its strengths. The largest correction I would ask for is that the authors refer uniformly to “gradient ascent” throughout the paper, since they apply their gradient-estimation framework to objective maximization, rather than the minimizing gradient descent normally used in much of machine learning.

Relation to Prior Work: While the authors situate their work relative to prior work in the Discussion section instead of earlier, they do so very clearly.

Reproducibility: Yes

Additional Feedback:

[Author Response · NeurIPS 2020]

We thank the reviewers (R1, R2, R3) for their detailed feedback. Key concerns are briefly addressed below.

**Novelty of learning rule and key contributions (R2):** While the final expression for our learning rule is superficially
similar to other tri-factor rules (as suggested by the name), ours is derived in a principled manner from a global objective.
Importantly, our solution is by construction designed for recurrent networks, unlike the references mentioned by R2,
which do not learn recurrent weights (note that we had originally cited two of them). The closest solution to ours at the
technical level is the REINFORCE algorithm (Williams, 1992), but the Hebbian term in that formulation is different
and produces subtly different experimental predictions:

• Our learning rule: $\Delta w_{ij} \propto \alpha(\mathbf{Dr}, \mathbf{s}) \left( r_i r_j - \langle r_i r_j \rangle_{p(\mathbf{r}|\mathbf{s})} \right) - 2\lambda_W w_{ij}$

• REINFORCE: $\Delta w_{ij} \propto \alpha(\mathbf{Dr}, \mathbf{s}) \sum_{t=0}^{T} f'(h_i(t))(r_i(t) - \bar{r}_i(t)) r_j(t) - 2\lambda_W w_{ij}$,

where $f(\cdot)$ is the activation function, $h_i$ is the pre-activation of neuron $i$, and $\bar{r}_i$ is the expected average activation for
that neuron. One notable difference is that learning is gated by deviations from the mean co-activation of the pre- and
post-synaptic neurons, and not by post-synaptic activity alone – which is unique to our solution and potentially testable.

More generally, to our knowledge, no previous work has used tri-factor learning rules to study the effects of task
constraints and intrinsic noise on the learned representations. Our derivation of the three-factor rule facilitates this goal,
but our main contribution is these novel analyses (as noted by R1).

**Disentangling sensory information from task (R3):** While one tends to think of sensory representations as being
determined exclusively by input statistics, with task constraints only affecting decoding/decision-making circuitry, there
is a substantial body of experimental evidence showing that early sensory cortices can change in a task-specific manner
in the presence of neuromodulation (Polley & Merzenich, 2006; Froemke & Schreiner 2007). Our results speak to these
experimental observations. More generally, the task specificity of the learned code will depend on several factors –
the set of tasks the system needs to perform, and various resource constraints (architecture, total number of neurons,
etc). We believe that generalizations of our circuit model should allow one to dissect the contribution of each of these
elements to the final representation.

**Symmetric weights (R1):** Having a proper energy significantly simplifies the analysis of how the noise is being
reshaped during learning, but is not strictly necessary for our framework. We are working on a generalization to
arbitrary weights, which results in qualitatively similar learning rules, with additional temporal integration via eligibility
traces. Also note that our current learning rule will naturally converge to symmetric weights, even for arbitrary synapse
initialization (cf. Kolen & Pollack, 1994), so symmetry is an emergent property of the network.

**Results clarifications (R1):** Prior manipulations were made throughout learning, not only at test time. The volume
fraction is defined as $VF(\mathbf{s}) = \frac{\det \mathbf{C}_D}{\det \mathbf{C}_{PCA}}$, where $\mathbf{C}_D$ is the covariance matrix of $\mathbf{r}$ projected onto the two output
dimensions, and $\mathbf{C}_{PCA}$ is the covariance matrix of $\mathbf{r}$ projected onto the first two principal components of the neural
activity for a fixed stimulus $\mathbf{s}$. In Fig. 2a, ellipses give 95% confidence range for network outputs, for a range of test
stimuli marked with corresponding black crosses.

"It is not clear in what way *intrinsic noise in the recurrent dynamics ... allows us to derive closed-form probabilistic*
*expressions for the objective function gradients.*": when we say that noise is necessary for our learning rule, we mean
that the probabilistic description of neural activity is essential for the derivation of the learning rule (Eq. 3), not that the
noise levels must be large. Intrinsic noise is what allows for this probabilistic description. In particular, in the absence
of noise, our learning rule produces weight updates of 0 (i.e., no learning).

**Intrinsic noise is not always bad (R3):** Increasing the magnitude of the noise is generally detrimental for performance,
at least in our setup (Fig. 3c). We thought that the network might learn to use its stochasticity to encode uncertainty, but
saw no evidence for this for our simple tasks. We'll rephrase the statement about noise being detrimental for encoding,
mentioning potential benefits of sampling for Bayesian computation.

*The network just plays as a filter of the noise in the framework of MSE (point estimate).* You are right that the output of
the network is a point estimate and not a probability. But there is a subtle point to be made here: although we have a
probabilistic formulation for the encoding model, the computational goals of the circuit are not explicitly Bayesian. The
task objectives are defined by marginalizing the prior input statistics and the intrinsic noise.

**Clarity and missing details (R1-3):** We will correct the typos, add the requested additional information and the
suggested references in the updated version. A Github code repository will also be provided to facilitate reproducibility.

[Meta-Review · NeurIPS 2020]

This paper proposes a stochastic recurrent neural network that builds up its local information representation through a learning rule based on Boltzmann machines, but weighted by a task-dependent objective function, forming a so-called tri-factor learning rule. The results show how the network depends on the tasks of regression and classification in terms of the distribution of the tuning curves, population-averaged activities, and dependence on stimulus priors. The paper then considered how noises are redistributed in the neural manifold such that task performance can be achieved. Reviewers were overall positively predisposed towards this submission. Strengths include the coherent derivation of the proposed learning rule and the thorough analysis of its properties. The main point of contention in the reviews, which was based by one reviewer, was whether the proposed learning rules are in fact novel. This reviewer noted that the proposed rule appeared very similar to the the rules in Eq (7) of reference 34 and Eq (16) in a paper by Legenstein et al. The AC recruited a 4th emergency reviewer, who also expressed a positive opinion of the paper. The AC also took a look at the discussed references to understand the degree of novelty of the proposed learning rule. This is not straightforward, since these rules are discussed in papers that are decades apart, and consider different settings and notation. From the author response, it is clear that the proposed rule is not the same as the rule in the Williams paper. It is also not clear to the AC that the proposed rule is the same as Eq (16) in Legenstein et al, which computes a covariance between the total synaptic input and the reward. Eq (7) in Ref 34 appears to compute a scalar variance between the firing rate and readout. Both equations rely on a low-pass-filtered time-average, which the proposed equations do not. The ACs best judgement is that, while it could be the case that these equations are in some respect equivalent to the proposed learning rule, any equivalence is not trivial. Based on the fact that the main claimed contribution is the application of this learning rule to the proposed recurrent network, the AC is inclined to say that acceptance is warranted. The AC would like to strongly request that the authors revise their manuscript to clearly and explicitly discuss in what ways the proposed rule is similar rules that have been previously considered in the literature, and in what way it is differs. This will strengthen the paper, since other readers will likely have a similar confusion.